# M³T2IBench: A Large-Scale Multi-Category, Multi-Instance, Multi-Relation Text-to-Image Benchmark

## ABSTRACT

Text-to-image models are known to struggle with generating images that perfectly align with textual prompts. Several previous studies have focused on evaluating image-text alignment in text-to-image generation. However, these evaluations either address overly simple scenarios, especially overlooking the difficulty of prompts with multiple different instances belonging to the same category, or they introduce metrics that do not correlate well with human evaluation. In this study, we introduce M³T2IBench, a large-scale, multi-category, multi-instance, multi-relation along with an object-detection-based evaluation metric, $AlignScore$, which aligns closely with human evaluation. Our findings reveal that current open-source text-to-image models perform poorly on this challenging benchmark. Additionally, we propose the Revise-Then-Enforce approach to enhance image-text alignment. This training-free post-editing method demonstrates improvements in image-text alignment across a broad range of diffusion models. [1]

## 1 INTRODUCTION

Text-to-Image (T2I) models have demonstrated impressive performance in generating high-quality, realistic images (Betker et al., 2023; Esser et al., 2024). Despite this success, T2I models continue to struggle with accurately interpreting and following user prompts. They may fail to generate objects with the correct number, attributes, or relationships (Li et al., 2024). However, assessing the alignment between text and generated image has remained a longstanding challenge.

There are generally three approaches to evaluating image-text alignment. The first approach involves using pretrained image-text models to generate an overall alignment score. CLIP Score (Hessel et al., 2021) is a widely used metric, while VQAScore (Lin et al., 2024) is an improved version of CLIP Score. However, these metrics have several limitations, including their inability to accurately reflect the true alignment between the image and the text (Li et al., 2024) and failing to provide explainable evaluation results.

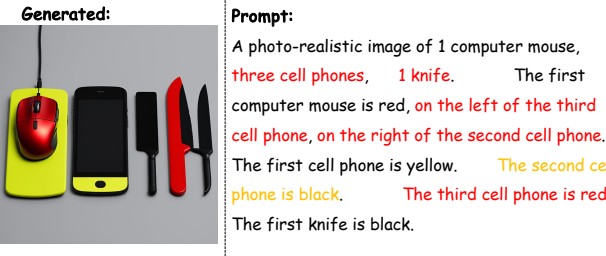

Figure 1: A failure case generated by Stable-Diffusion-3. The right is the prompt and the left is the generated image. We label the parts incorrectly generated in red and questionable parts in yellow.

---

[1]Our code and data has been released in supplementary material and will be made publicly available after the paper is accepted.

The second approach decomposes text prompts into components and analyzes image-text alignment based on these components, such as through question generation and answering (QG/A) methods (Hu et al., 2023; Cho et al., 2023). This approach is capable of handling nearly any type of text prompt. However, a major challenge is that it is difficult to decompose a text prompt into disjoint, yet equally relevant components (Cho et al., 2023). This is even more severe when there are multiple instances belonging to the same category in the prompt, where QG/A methods struggle to correctly identify a certain instance in the generated question.

The third approach focuses on constructing structured inputs and evaluating the generated results based on these inputs using corresponding tools. Although it cannot handle arbitrary prompts, this method provides more reliable and reproducible evaluation results for the constructed prompts and offers a better reflection of model performance. GenEval (Ghosh et al., 2023) is a benchmark that follows this approach. However, GenEval is limited to simple cases, such as those involving only one or two object categories.

In this work, we adopt the third approach to provide a more reliable evaluation of text-to-image generation, but aiming to extend the evaluation to more complex scenarios. To this end, we propose **M$^3$T2IBench**, a large-scale **M**ulti-Category, **M**ulti-Instance, **M**ulti-Relation text-to-image benchmark. This benchmark includes prompts with multiple categories, multiple instances within each category, attribute (color) assigned to each instance, and various spatial relations between instances. The benchmark is organized in a structured format and translated into natural language using a carefully designed template, ensuring both the quality of the prompts and ease of evaluation. Additionally, we introduce an evaluation metric, $AlignScore$, to assess image-text alignment in this complex setting without introducing expensive closed-source models. Specifically, we propose a method to match the generated object instances in the images with the instances described in the prompt, facilitating accurate metric calculation. We show that $AlignScore$ correlates with human evaluation well.

Using our benchmark, we evaluate a wide range of T2I models and reveal that all models occasionally fail to accurately follow prompts, particularly as the complexity of the prompts increases, like the example shown in Figure 1. We further reveal several challenges models face when prompts get complicated.

Furthermore, we propose a "Revise-Then-Enforce" method, a training-free post-editing approach designed to improve image-text alignment. The key idea behind this method is to identify the incorrectly generated parts of the image and then construct a pair of additional prompts to guide the model away from making similar mistakes. This approach is applicable to almost all T2I diffusion models and results in performance improvements across all models we tested.

To summarize, the contributions of our work are listed as follows:

- We propose M$^3$T2IBench, a **M**ulti-Category, **M**ulti-Instance, **M**ulti-Relation text-to-image benchmark with a metric named $AlignScore$ which aligns with human evaluation much better.
- We evaluate a wide range of T2I models and reveal the challenges of generating well-aligned images with complex text prompts.
- We propose a "Revise-Then-Enforce" method, a training-free post-editing method to make the generated image align more closely with the text prompt.

## 2 RELATED WORKS

### 2.1 TEXT-TO-IMAGE DIFFUSION MODELS

(Ho et al., 2020) first introduced DDPM, serving as the foundation for diffusion models. (Dhariwal & Nichol, 2021) proposed classifier guidance and (Ho & Salimans, 2022) proposed classifier-free guidance. (Rombach et al., 2022) proposed denoising in latent space.

Later studies introduced more practical text-to-image diffusion models, including open-source ones: Imagen (Ho et al., 2022), Stable-Diffusion-XL (Podell et al., 2023), DiT(Peebles & Xie, 2023), Stable-Diffusion-3(Esser et al., 2024), Pixart-$\Sigma$ (Chen et al., 2024b) and closed-source ones: DALLE-2 (Ramesh et al., 2022) and DALLE-3 (Betker et al., 2023).

Table 1: Comparison of existing benchmarks and M³T2IBench. ✓ means the benchmark satisfies the corresponding property and ✗ refers to benchmark not satisfying the corresponding property. * denotes only complex prompts like ours are considered.

| Benchmark | Size | Maximum prompt length | Maximum relation number | Maximum instance number | Different Instances of the Same Category | Structured Data |
|---|---|---|---|---|---|---|
| GenEval | 100* | $\leq 20$ | 1 | 2 | ✗ | ✓ |
| T2I-CompBench | 1000* | 32 | 3 | 3 | ✗ | ✗ |
| GenAI-Bench | 1600 | 42 | 2 | 4 | ✓ | ✗ |
| ConceptMix | 2100 | 50 | 3 | 3 | ✗ | ✓ |
| **M³T2IBench** (ours) | **10000** | **78** | **6** | **5** | ✓ | ✓ |

## 2.2 IMAGE-TEXT ALIGNMENT

CLIP-Score (Radford et al., 2021; Hessel et al., 2021) evaluates image-text alignment by computing the cosine similarity of CLIP embeddings. VQAScore (Lin et al., 2024) queries a VQA model to determine if the image corresponds to the prompt, using the "Yes" logit as the metric. T2I-CompBench (Huang et al., 2023) leverages MiniGPT-4 (Zhu et al., 2023) CoT to generate an alignment score. These metrics do not provide fine-grained evaluation and are known to sometimes yield less reliable results (Li et al., 2024; Xiong et al., 2023).

Decomposition-based metrics break down text prompts into smaller components and assess the accuracy of each part. TIFA (Hu et al., 2023) generates visual questions and uses a VQA model to verify the correctness of each component. T2I-CompBench (Huang et al., 2023) employs Blip-VQA (Li et al., 2022), while DavidSceneGraph (Cho et al., 2023) and VQ² (Yarom et al., 2024) are similar to TIFA. MHalu-Bench (Chen et al., 2024c) builds a pipeline of tools to check the correctness of each component directly. However, all decomposition-based methods face the challenge of correctly decomposing the prompt into questions and correctly answering these questions. ConceptMix(Wu et al., 2024a) is a more complicated benchmark, but it does not contain prompts with different instances of the same category and relies on GPT-4o for evaluation, making the consistency of the evaluation results questionable.

Some other benchmarks include (Gokhale et al., 2023; Patel et al., 2024), yet they are too simple. GenEval (Ghosh et al., 2023) is a unique benchmark that relies solely on an object detector and CLIP (Radford et al., 2021) for evaluation, enhancing evaluation stability. However, GenEval is limited to simple prompts, and more complex scenarios remain unaddressed.

Several methods have been proposed to improve image-text alignment. (Liu et al., 2022) uses conjunction and negation prompts to combine different components. Attend-and-Excite (Chefer et al., 2023) and (Wang et al., 2024) leverage attention maps to enhance alignment. LayoutGPT (Feng et al., 2023) employs GPT-4 (OpenAI et al., 2023) to generate prompt layouts. GLIGEN (Li et al., 2023) and SG-Adapter (Shen et al., 2024) are training-based methods.

## 3 M³T2IBENCH: CONSTRUCTION AND EVALUATION

In this section, we present the construction framework of our benchmark and the corresponding evaluation metrics. First, we highlight the key differences between our proposed benchmark and existing works as shown in Table 1. Our benchmark is large in scale and includes prompts that cover complex scenarios, incorporating multiple categories, instances, and relationships. Additionally, it includes structured data for each prompt to enable more effective evaluation as illustrated in Figure 2. An overview of our benchmark and metrics is shown in Figure 3.

## 3.1 BENCHMARK CONSTRUCTION

To accurately evaluate image-text alignment, we construct our benchmark in an object-centered manner, similar to (Ghosh et al., 2023). [2] Each data point in our benchmark consists of structured

---

[2] We use "object category" (or "category") to refer to the type of an object, and "object instance" (or "instance") to refer to a specific object.

**Structured Data:**
[*total_number*]: 4
[*bench*] : (id:0, white, left: (boat, 0))
(id:1, black) (id:2, red)
[*boat*]: (id: 0, green)
**Prompt:** A photo-realistic image of three bench, one boat. The first bench is white, on the left of the first boat. The second bench is black. The third bench is red. The first boat is green.

Figure 2: Example of data in our benchmark.

**Step 1: Benchmark Construction**

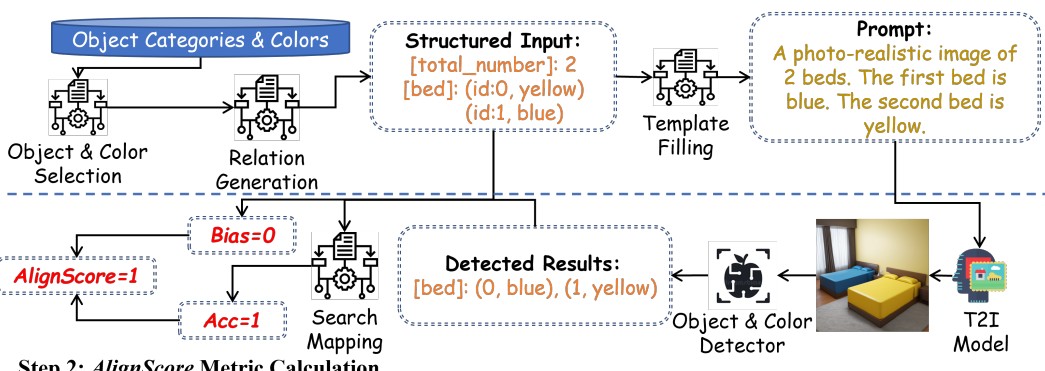

**Step 2: *AlignScore* Metric Calculation**

Figure 3: Overview of our benchmark construction and metric calculation.

information, including several object instances that may belong to the same or different categories, attributes (color) assigned to each instance, and the spatial relationships between them. A textual prompt is then generated based on this structured data. To construct this benchmark, we introduce three steps: object and color selection, relation generation, template filling.

**Object and Color Selection**    Following (Ghosh et al., 2023), we initially use the 80 categories labeled by MSCOCO (Lin et al., 2014) to ensure compatibility with popular pretrained object detectors, along with the color categories defined by (Berlin & Kay, 1991). When generating a data point, we randomly select several object categories and assign a random number of instances to each category. Each object instance is then assigned a color.

We choose color as the attribute for discussion primarily to reduce ambiguity in evaluation. As a comparison, prompts in GenAI-Bench (Li et al., 2024) often include attributes that are somewhat subjective (e.g., "fresh," "majestically"), making accurate evaluation of image-text alignment highly challenging. Details on our object and color selection can be found in Appendix A.

**Relation Generation**    After generating object instances, we then generate spatial relationships between them. For this research, we select four relationships: above, below, left, and right. We randomly assign a spatial relationship to some instance pairs.

Unlike (Ghosh et al., 2023), this procedure can generate multiple spatial relationships within a single prompt, making it necessary to prevent the formation of cycles (e.g., A is above B, B is above C, and C is above A). To address this, we apply a topological sort to detect and eliminate such cycles. For further details, please refer to Appendix A.

**Template Filling**    To construct a prompt based on the generated object instances, attributes, and relations, we design a template in the following format:

```
A photo-realistic image of [n_k] [category_k].    The [i-th] [category_k]
is [color_ik], [relation_ij].
```

We use ids to identify object instances in the constructed prompt. An example of our constructed prompt can be found in Figure 2.

Unlike T2I-CompBench(Complex split) (Huang et al., 2023), we use a template instead of ChatGPT to ensure the quality of our prompts. As discussed in (Li et al., 2024), prompts generated by Chat-GPT often include subjective descriptions, such as "a stunning centerpiece of beauty," which can complicate evaluation.

**Benchmark Statistics** We have constructed a benchmark containing 10,000 data points, which is larger than most existing T2I alignment benchmarks. Given the complexity of our scenarios and the diversity of generated prompts, we believe that a larger benchmark will better enhance the robustness of evaluation. Some details of our benchmark, including the distribution of data based on the total number of instances and the number of instances belonging to the same category, are shown in Figure 4.

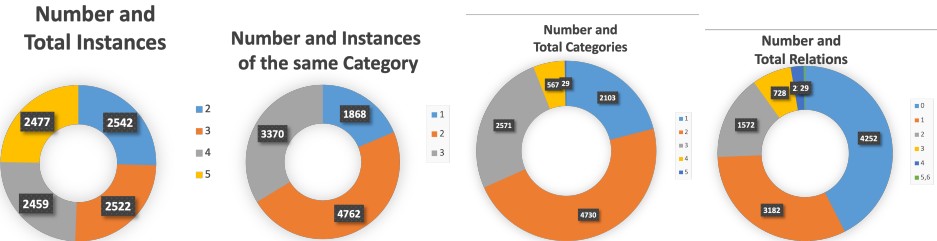

Figure 4: Statistics of our benchmark.

As can be seen, since our benchmark is relatively large, generally each split of our benchmark contains over 100 prompts, ensuring the stability of evaluation while allowing a more diverse prompt.

## 3.2 EVALUATION METRICS

Since our benchmark includes structured input, we can employ an evaluation framework that directly assesses image-text alignment for each part of the prompt, enhancing the trustworthiness of the evaluation. Additionally, unlike (Ghosh et al., 2023), which uses a simple correct/incorrect classification for final evaluation, we utilize fine-grained metrics to provide a more comprehensive assessment of the model's generation performance. Intuitively, evaluating the alignment in our benchmark consists of two aspects: **checking of number of generated object instances, namely "bias"**, and **checking the accuracy of generated attributes and relations, namely "accuracy"**.

Consider $N$ object instances $\{o_1, ..., o_N\}$ mentioned in the prompt , each instance $o_i$ is attributed with color $c_i$, and the relation between instance $o_i$ and instance $o_j$ is $r_{ij}$ (if pair $(o_i, o_j)$ is assigned a spatial relationship). We use an **object detector** to identify instances in the generated image, denoted as $\{o'_1, ..., o'_M\}$. A **color detector** is then applied to detect the color of each instance, denoted as $\{c'_1, ..., c'_M\}$ correspondingly. Once the instances are detected with bounding boxes, the spatial relations can be determined by comparing the bounding boxes. The detected relation between $(o'_i, o'_j)$ pair is denoted as $r'_{ij}$.

To calculate bias, we can simply use

$$Bias = \sum_{k=1}^{K} |n_k - m_k| \tag{1}$$

$K$ is the total number of categories in the prompt, $n_k$ is the number of instances belonging to category $k$ required by the prompt, and $m_k$ is the number of instances belonging to category $k$ detected in the generated image.

To assess accuracy, formally, consider the case where $M \geq N$ to illustrate our evaluation metric. If $M < N$, we can add $N - M$ "blank" instances $\{o'_{M+1}, ..., o'_N\}$ to the detection result, effectively converting the case to $M \geq N$.

Denote $[N] = \{1, ..., N\}$, when $M \geq N$, we aim to find an injection [3] $f : [N] \to [M]$, such that $\forall i \in [N], o'_{f(i)}$ either belongs to the same category as $o_i$ or is a blank object. The function $f$ represents a matching between the instances detected in the image and those mentioned in the prompt. Once the matching is determined, we can directly calculate the accuracy of color and relation as follows:

$$Acc = \frac{1}{|Z|}(\sum_{i=1}^{N}\mathbb{I}[c_i = c'_{f(i)}] + \sum_{i=1}^{N}\sum_{j=1}^{N}\mathbb{I}[r_{ij} = r'_{f(i)f(j)}]) \tag{2}$$

where $|Z|$ is a normalizing factor and $r_{ij}$ is calculated only for instance pairs with a given relation, with **each pair being evaluated only once**, $\mathbb{I}[x] = 1$ if $x$ is True otherwise $\mathbb{I}[x] = 0$.

Naturally, if $o'_{f(i)}$ is a blank object, $\mathbb{I}[c_i = c'_{f(i)}] = 0$ and $\mathbb{I}[r_{ij} = r'_{f(i)f(j)}] = \mathbb{I}[r_{ji} = r'_{f(j)f(i)}] = 0$.

**The key challenge in our evaluation arises from the possibility of multiple instances belonging to the same category, leaving the possibility of there being multiple possible matching $f$, and different choices of $f$ may lead to different calculated $Acc$.** Therefore, our goal is to find the most reasonable $f$. (Note that different $f$ does not affect $Bias$)

Intuitively, the most reasonable $f$ should be the one that yields the highest $Acc$. Otherwise, we can simply adjust $f$ to achieve a higher $Acc$, implying that a low $Acc$ may stem from incorrect selection of $f$ rather than a failure in generation. This suggests that only the highest $Acc$ among all possible $f$ is meaningful for evaluation.

To identify this $f$, we propose an exhaustive search algorithm that evaluates all possible $f$, calculates the $Acc$ for each $f$, and selects the one with the highest $Acc$. If multiple $f$ yield the same highest $Acc$, we randomly select one, as this does not affect the final evaluation metric. The highest $Acc$ is then used as the $Acc$ measurement of our benchmark.

Furthermore, two separate metrics can sometimes make it difficult to directly compare model performance. To address this, we propose a simple combined metric $AlignScore$, defined as follows:

$$AlignScore = \frac{1}{2}(Acc + \frac{1}{Bias + 1}) \tag{3}$$

Following (Ghosh et al., 2023), the calculation of $Acc$ and $Bias$ can be performed through an object detector and a color detector. We further conduct a human evaluation and reveal that our metric $AlignScore$ correlates well with human evaluation, with a Pearson $r = 0.6711$ and Kendall $\tau = 0.5348$, greatly surpassing previous popular metrics including CLIPScore (Hessel et al., 2021), VQAScore (Li et al., 2024) and DSGScore (Cho et al., 2023). We present details about metric calculation, human evaluation and more discussion Appendix A.

## 4 METHOD: REVISE-THEN-ENFORCE

In this section, we introduce a simple yet effective training-free post-editing method to generate images better aligning with prompts in diffusion T2I models.

### 4.1 PRELIMINARIES

Most T2I diffusion models use classifier-free guidance (CFG) to condition on the given text. For simplicity, the prediction of a denoising diffusion model can be represented as $z(x_t, c)$ at timestep $t$ at timestep $t$, conditioned on $c$ and $z(x_t, \phi)$ at timestep $t$ without conditioning. The final score used by CFG is formulated as:

$$z_t = z(x_t, \phi) + w(z(x_t, c) - z(x_t, \phi)) \tag{4}$$

---

[3]Injection: a function $f : \mathcal{X} \to \mathcal{Y}$ satisfying $\forall x_1 \neq x_2, f(x_1) \neq f(x_2)$

where $w$ is a hyper-parameter.

In word vectors, denote $e(s)$ being the word embedding of a word $s$, then we can perform semantic transforms with vector calculation (Mikolov et al., 2013), like a classic example:

$$e(king) + (e(woman) - e(man)) \approx e(queen) \tag{5}$$

### 4.2 REVISE-THEN-ENFORCE

Inspired by word vectors, we hypothesize that a similar phenomenon occurs in the denoising process, as the formulation in (4) mirrors that in (5). Specifically, if we formulate CFG as follows:

$$z_t = z(x_t, \phi) + w(z(x_t, c_0) - z(x_t, \phi)) + w^{'}(z(x_t, c_1) - z(x_t, c_2)) \tag{6}$$

The generated result should be a **semantic combination** of $c_0$ and $c_1 - c_2$. For example, like the example in Equation 5, if $c_0$ is "king", $c_1$ is "woman" and $c_2$ is "man", the generation result should be similar to "queen" instead of "a king and a woman". An example is in Appendix C.

Therefore, if we can identify the failure parts in image-text alignment, we can determine how these parts should be generated as $c_1$, and how these parts are currently incorrectly generated as $c_2$. This guides the model to avoid these failures and improve alignment with the text prompt. The key insight is to construct $c_1$ and $c_2$ as paired prompts with related but distinct semantics, as shown in Equation 5, to guide the generation more effectively without introducing other interruptions.

To summarize, our method consists of two parts: **Revise**: identifying the failed parts of the prompt and how they are currently generated, and **Enforce**: specifying how the failed parts should be generated in $c_1$ and how they are currently incorrectly generated in $c_2$. In our benchmark, the **Revise** procedure is automatically performed during the evaluation metric calculation. Since our method involves identifying the failed parts in an already generated image, it is actually a post-editing method.

We describe the detailed process of our method as follows: Given a prompt $p$, sample a noise prior $x$ and use formulation $4(c = p)$ to generate an image and identify the misaligned part between the generated image and prompt $p$. Then we form how these misaligned parts should be generated into $c_1$, how these misaligned parts are currently generated into $c_2$, maintain noise prior $x$ unchanged and use formulation $6(c = p)$ to generate the final image output.

It is worth mentioning that, though the formulation 6 comes from the insight of word vectors naturally, there are other variants of this method worth discussing. First of all, $c_2$ in 6 appears in a similar position with a common concept "negative prompt" in diffusion models, so we remove $c_1$ and view $c_2$ as a negative prompt, thus formulating the denoising process as:

$$z_t = z(x_t, \phi) + w(z(x_t, c) - z(x_t, c_2)) \tag{7}$$

which is similar to that proposed in (Desai & Vasconcelos, 2024), and we name it **Negative Prompt** .

We also would like to investigate whether $c_2$ is useful, so we provide another formulation:

$$z_t = z(x_t, \phi) + w(z(x_t, c) - z(x_t, \phi)) + w^{'}(z(x_t, c_1) - z(x_t, \phi)) \tag{8}$$

which is similar to that proposed in (Wang et al., 2023), and we name this variant **Positive Prompt**.

## 5 EVALUATION RESULTS AND ANALYSIS

### 5.1 EVALUATION SETUP

For base models, we select six open-source models for evaluation: Stable-Diffusion-3.5-Medium(SD3.5) (Esser et al., 2024), Stable-Diffusion-3.5-Large-Turbo (Esser et al., 2024), FLUX.1-dev (Labs, 2024), Stable-Diffusion-3 (SD3) (Esser et al., 2024), Stable-Diffusion-1.5 (SD1.5) (Rombach et al., 2022), PixArt-$\Sigma$-XL (Chen et al., 2024b). These models vary in size, architecture, and training data, making them representative. We also include DALLE-3 (Betker

et al., 2023) as an example of closed-source models. Due to resource constraints, we only test 100 samples on DALLE-3.

Table 2: Results of Different T2I models and methods on our benchmark. $Acc$, $AlignScore$, VQAScore are multiplied by 100 for better presentation. DALLE-3 is tested on a subset of our benchmark with 100 prompts. We label the best result within the same base model in **bold** and the best result among all methods in underline.

| Method Name | Base Model | $AlignScore(\uparrow)$ | $Acc(\uparrow)$ | $Bias(\downarrow)$ | VQAScore $(\uparrow)$ |
|---|---|---|---|---|---|
| Base | SD1.5 | 26.7 | 28.2 | 2.98 | 36.4 |
| A&E | | 26.8 | 28.0 | 2.90 | 36.6 |
| Composable | | 26.5 | 28.7 | 3.09 | **37.6** |
| R&E(Ours) | | **27.5** | **29.3** | **2.89** | 37.5 |
| Base | Pixart-$\Sigma$ | 42.0 | 53.1 | 2.24 | 51.8 |
| R&E(Ours) | | **44.1** | **55.6** | **2.09** | **52.9** |
| Base | SD3 | 53.3 | 66.9 | 1.52 | 58.0 |
| Negative Prompt | | 50.5 | 63.4 | 1.66 | **59.6** |
| Positive Prompt | | 53.3 | 67.0 | 1.53 | 58.0 |
| R&E(Ours) | | **55.1** | **69.2** | **1.43** | 58.5 |
| Base | SD3.5 | 52.0 | 62.9 | 1.43 | 58.8 |
| R&E(Ours) | | **53.1** | **65.4** | **1.37** | **59.0** |
| Base | SD3.5-Large-Turbo | 50.8 | 61.9 | 1.52 | 71.3 |
| Base | FLUX.1-dev | 54.8 | 65.7 | 1.27 | 56.3 |
| Base | DALLE-3* | 51.0 | 60.5 | 1.41 | 49.0 |

In addition to these base models, we also include some popular training-free methods for more comprehensive evaluation, including Composable Diffusion (Composable) (Liu et al., 2022), Attend-and-Excite (A&E) (Chefer et al., 2023). Since these methods are implemented on certain base models (SD1.5), we only report their results on SD1.5. We also report the results of our proposed method, Revise-Then-Enforce (R&E), on all possible models to show the wide applicability of our method. Results of the two variants of our method based on SD3 are also reported.

In addition to our proposed metrics, we also introduce VQAScore (Lin et al., 2024) as a metric because it is commonly used. The inference hyper-parameters are selected following their default settings. We generate only one image per prompt. Details about the experiment setup are listed in Appendix B.

### 5.2 Evaluation Results and Analysis

The evaluation results are presented in Table 2. As shown, the performance of all models on our benchmark is unsatisfactory, with average accuracy falling below 70% and average bias exceeding 1, which is clearly unacceptable.

**Overall Analysis** Stable-Diffusion-3 performs the best in terms of accuracy, while FLUX.1-dev excels in bias and overall $AlignScore$. This also underscores that accuracy and bias are not always aligned, highlighting the importance of considering both during evaluation. Stable-Diffusion-3.5-Large-Turbo shows abnormally highest VQAScore with other metrics remaining low, so we attribute this phenomenon to the flaw of VQAScore. Stable-Diffusion-1.5 performs poorly on both aspects, demonstrating that newer, larger models are generally more effective at image-text alignment.

The closed-source model, DALLE-3, also does not perform well on our benchmark, which actually aligns with the observation in (Esser et al., 2024). This observation further highlights the necessity of our benchmark.

In the following discussions, we highlight some of the challenges introduced by our benchmark, including multi-category, multi-instance, and multi-relation scenarios. We fix the total number of object instances at 4 and present the model performance ($Acc$ and $Bias$) in Figure 5.

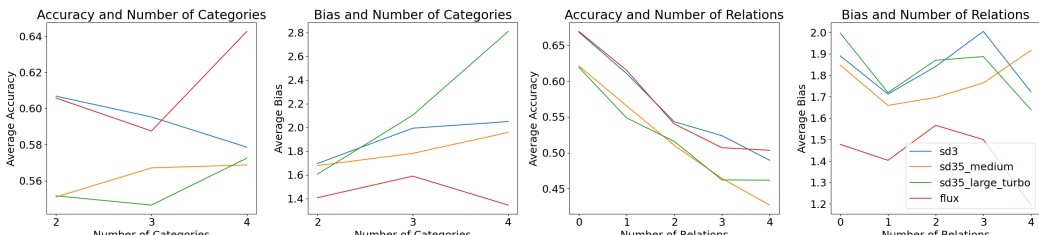

Figure 5: Average $Acc$ and $Bias$ under different number of categories or number of relations. Total number of instances is fixed as $4$.

CHALLENGE 1: MULTI-INSTANCE TENDS TO DECREASE $Acc$  As shown in the first image of Figure 5, in most cases, models experience a decrease in $Acc$ as more instances of the same category (i.e., fewer categories) are introduced. We hypothesize that distinguishing attributes and relations between instances of the same category is challenging, underscoring the difficulty of generating accurate attributes and relations when multiple instances of the same category are presented.

CHALLENGE 2: MULTI-CATEGORY TENDS TO INCREASE $Bias$  Similarly, as shown in the second image of Figure 5, models tend to exhibit an increase in $Bias$ as more categories are introduced in the prompt. We attribute this phenomenon to models making independent counting errors for each category. Consequently, the more categories there are, the more mistakes the model is likely to make. This underscores the challenge of generating the correct number of instances when multiple categories are involved in a prompt.

CHALLENGE 3: MULTI-RELATION TENDS TO DECREASE $Acc$  As shown in the third image of Figure 5, introducing more relations into a prompt leads to a significant decrease in $Acc$, with the decline almost being linear. This suggests that generating correct relations is particularly challenging, and the models appear to "randomly" fail on certain aspects of the relations. Interestingly, the addition of multiple relations does not necessarily result in a higher $Bias$. The impact of multi-relations on $Bias$ seems to be somewhat random and warrants further investigation.

We present more detailed analysis and case study in Appendix B.

**Analysis of Our Method**  As shown in the results, the performance of all models improves when **Revise-Then-Enforce** is applied, while previous methods do not provide notable improvement. Our method not only enhances performance on our metrics but also shows improvement on VQAScore, further demonstrating its effectiveness. Both stronger and weaker models benefit from this approach, highlighting the generalization ability of our method.

ABLATION STUDY  As can be seen, only our formulation 6 achieves improvement on all four metrics, while other formulations show little improvement or even lead to performance drop. Though formulation 7 shows improvement in VQAScore, it shows a severe performance drop on other metrics, so we attribute this improvement to the flaw of VQAScore.

This result clearly demonstrates the usefulness of both $c_1$ and $c_2$. It shows that only using paired prompts is effective to improve image-text alignment, while using either only positive ($c_1$) or negative ($c_2$) prompt is insufficient, highlighting the superiority of our formulation 6. We present a case of our method in Appendix C.

## 6 CONCLUSION

In this paper, we introduce a new benchmark, M³T2IBench, a large-scale multi-category, multi-instance, multi-relation text-to-image generation benchmark, along with fine-grained metrics $Acc, Bias, AlignScore$ to evaluate image-text alignment. We evaluate several T2I models and find that all of them perform poorly on our benchmark, particularly on the more complex parts. To address this, we propose the **Revise-Then-Enforce** method, a training-free post-editing technique, to effectively improve image-text alignment.

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

## A  BENCHMARK DETAILS

### A.1  OBJECT AND COLOR SELECTION

We find naive object and color selection imperfect, as some color-object pairs are either unrealistic or ambiguous, making both generation and evaluation difficult.

We invite two annotators to annotate the possible colors of an object category and select the intersection of their annotations as the final result. If an object category is not annotated with any possible colors, this category is then removed from our benchmark construction.

After this annotation, 66 out of 80 categories remain with each category annotated with 1 to 7 possible colors. All colors lie in the set {green, red, yellow, brown, black, white, blue}.

## A.2 Relation Generation

We construct relations between instance pairs in the following way. Each instance pair $(o_i, o_j)$ is given at most one spatial relation, and if the pair is given a relation, it is given "left" ($o_i$ is on the left of $o_j$), "right" ($o_i$ is on the right of $o_j$), "above" ($o_i$ is above $o_j$) and "below"($o_i$ is below $o_j$) under equal probability.

To be more detailed, we select a hyper-parameter $p = 0.05$, and generate relations "left" "right" "above" "below" with equal probability $p$. Therefore the probability of there being no relation between a pair of instance is $1 - p$.

The introduction of randomness help introduce the diversity of our benchmark and make it possible to observe model performance under different settings.

**Topological Sort**   As discussed, there can be multiple relations in a single prompt, thus, rings should be avoided (e.g., A is above B, B is above C, C is above A) to ensure the validity of the prompt. We use the classical algorithm of checking rings in a graph, topological sort. Note that actually we run the topological sort twice, once for checking possible rings caused by the above/below relations, the other for checking possible rings caused by the left/right relations. Take checking possible rings caused by left/right relation as an example, the topological sort algorithm is shown in Algorithm 1.

## A.3 Details about Metric Calculation

We generally follow GenEval (Ghosh et al., 2023) when performing object detection, color detection and relation detection. Specifically, we use Mask2Former [4] with Swin-S backbone available from the MMDetection toolbox from OpenMMLab (Chen et al., 2019) [5].

The Mask2Former generates bounding boxes, segmentations and confidence of detected object instances. We first conduct a non-maximal suppression to avoid detecting the same object instance more than once. We remove the bounding boxes with confidence less than $0.3$. We then investigate the bounding boxes from larger confidence to smaller. If a bounding box has an IoU over $0.9$ with another bounding box with higher confidence and the same category, this bounding box is removed as it may belong to an instance already detected by another bounding box. Both thresholds remain the same as (Ghosh et al., 2023). We also remove bounding boxes with one side less than $5$ pixels, since we believe these too small detection results come from noise instead of a real object.

For color detection, following the best practice from (Ghosh et al., 2023), when performing color detection of a detected instance, we first crop the image to the bounding box and then mask out the background and replace it with grey. Both steps are conducted to remove distraction (unrelated objects or background). We perform color classification by prompting CLIP (Radford et al., 2021) also like (Ghosh et al., 2023) [6]. Specifically, we use two prompt templates:

- The color of [Object] in this photo is [Color].
- The [Object] in this photo is [Color]-colored.

where [Object] is replaced by the category of the detected instance and [Color] is replaced with the name of the color. We form all possible colors into the prompts above and calculate cosine similarity with the cropped and masked image. The color corresponding to the prompt with the highest cosine similarity is selected as the detected color.

For relation detection, still following (Ghosh et al., 2023), consider two object instances $o_1, o_2$ with bounding boxes with size $(w_1, h_1), (w_2, h_2)$ and centered at $(x_1, y_1), (x_2, y_2)$, the relation can be determined as:

---

[4]`https://download.openmmlab.com/mmdetection/v2.0/mask2former/`
`mask2former_swin-s-p4-w7-224_lsj_8x2_50e_coco/mask2former_`
`swin-s-p4-w7-224_lsj_8x2_50e_coco_20220504_001756-743b7d99.pth`
[5]`https://github.com/open-mmlab/mmdetection`
[6]The model used is CLIP-ViT-L-14, checkpoint from `https://huggingface.co/openai/`
`clip-vit-large-patch14`

**Algorithm 1** Topological Sort

1: Input instances $\{o_1, ..., o_N\}$, relation set $R = \{r_{ij}\}$
2: Initialize $IND \leftarrow (0, ..., 0)$ as an vector with length $N$ filled with 0
3: **for** $r_{ij} \in R$ **do**
4:    **if** $r_{ij}$ is "left" **then**
5:       $IND[j] \leftarrow IND[j] + 1$
6:    **else if** $r_{ij}$ is "right" **then**
7:       $IND[i] \leftarrow IND[i] + 1$
8:    **end if**
9: **end for**
10: Initialize $Q = \{\}, S = \{\}$
11: **for** $i = 1, ..., N$ **do**
12:    **if** $IND[i] = 0$ **then**
13:       $Q \leftarrow Q \cup \{i\}, S \leftarrow S \cup \{i\}$
14:    **end if**
15: **end for**
16: **while** $|Q| > 0$ **do**
17:    Select $q \in Q$
18:    $Q \leftarrow Q - \{q\}$
19:    **for** $r_{qj} \in R$ **do**
20:       **if** $r_{qj}$ is "left" **then**
21:          $IND[j] \leftarrow IND[j] - 1$
22:          **if** $IND[j] = 0$ **then**
23:             $Q \leftarrow Q \cup \{j\}, S \leftarrow S \cup \{j\}$
24:          **end if**
25:       **end if**
26:    **end for**
27:    **for** $r_{iq} \in R$ **do**
28:       **if** $r_{iq}$ is "right" **then**
29:          $IND[i] \leftarrow IND[i] - 1$
30:          **if** $IND[i] = 0$ **then**
31:             $Q \leftarrow Q \cup \{i\}, S \leftarrow S \cup \{i\}$
32:          **end if**
33:       **end if**
34:    **end for**
35: **end while**
36: **if** $|S| = N$ **then**
37:    Return: No ring.
38: **else**
39:    Return: Ring.
40: **end if**

---

**Algorithm 2** Search Mapping

---

1: Input instances$\{o_1, ..., o_N\}$, corresponding color $\{c_1, ..., c_N\}$, relation set $R = \{r_{ij}\}$ mentioned in the prompt.
2: Input instances$\{o'_1, ..., o'_M\}$, corresponding color $\{c'_1, ..., c'_M\}$, relation set $R = \{r'_{ij}\}$ detected in the generated image.
3: Initialize Maximum $Acc$ $acc \leftarrow 0$
4: **for** all possible $f : [N] \rightarrow [M]$ **do**
5: $\quad c\_acc \leftarrow \frac{1}{|Z|}(\sum_{i=1}^{N} \mathbb{I}[c_i = c'_{f(i)}] + \sum_{i=1}^{N} \sum_{j=1}^{N} \mathbb{I}[r_{ij} = r'_{f(i)f(j)}])$
6: $\quad$ **if** $c\_acc > acc$ **then**
7: $\quad\quad acc = c\_acc, f_{best} = f$
8: $\quad$ **end if**
9: **end for**
10: Reture: $acc, f_{best}$

---

- $x_2 > x_1 + c(w_1 + w_2) \Rightarrow o_2$ is on the right of $o_1$

- $x_2 < x_1 - c(w_1 + w_2) \Rightarrow o_2$ is on the left of $o_1$

- $y_2 > y_1 + c(h_1 + h_2) \Rightarrow o_2$ is below $o_1$

- $y_2 < y_1 - c(h_1 + h_2) \Rightarrow o_2$ is above $o_1$

$c$ is a hyper-parameter for indicating that instances should be offset with each other to perceive spatial relations. $c = 0.1$ is used in our research.

We present mapping search algorithm in Algorithm 2.

The validity of the evaluation framework mentioned above has already been proved by (Ghosh et al., 2023). We conduct a human evaluation in the previous section to further validate the trustworthiness of our evaluation result.

## A.4 HUMAN EVALUATION

We conduct human evaluation generally following the protocol provided in (Otani et al., 2023; Li et al., 2024). We hire two human annotators to collect 1-5 Likert scale human ratings for image-text alignment. Both annotators are college students, showing enough capability to take such a task. The rating criteria is given following (Li et al., 2024):

---

**Human Annotation Criteria**

How well does the image match the description?
1. Does not match at all.
2. Has significant discrepancies.
3. Has several minor discrepancies.
4. Has a few minor discrepancies.
5. Matches exactly.

---

Figure 6: Criteria of human annotation.

We generate 500 samples using the models evaluated in our experiments. The annotators achieve a Krippendorff's alpha (Krippendorff, 2011) of 0.92, indicating a high level of agreement (Li et al., 2024). The average rating of the two annotators is used as the final human rating. We compute Pearson's $r$, Spearman's $\rho$ and Kendall's $\tau$ scores to measure the consistency between our metric and human evaluation. For comparison, we also evaluate using CLIP Score (Hessel et al., 2021), VQAScore (Lin et al., 2024), 3-in-1 Score and mGPT-CoT (Huang et al., 2023), and DSG (Cho et al., 2023). These metrics cover different types.

Table 3: Consistency with human evaluation.

| Metric | Pearson($r$) | Spearman ($\rho$) | Kendall($\tau$) |
|---|---|---|---|
| mGPT-CoT | 0.0514 | 0.0541 | 0.0408 |
| CLIP Score | 0.2296 | 0.2255 | 0.1594 |
| 3-in-1 Score | 0.2341 | 0.2304 | 0.1627 |
| DSG | 0.3713 | 0.3597 | 0.2634 |
| VQAScore | 0.4022 | 0.4008 | 0.2975 |
| $AlignScore$ (ours) | **0.6711** | **0.6679** | **0.5348** |

As can be seen from the results, our $AlignScore$ achieves the highest consistency with human evaluation. CLIP Score and 3-in-1 score perform poorly, while mGPT-CoT performs even worse, which is a little bit surprising. DSG and VQAScore align with human evaluation better, yet their alignment is still not satisfying. Specifically, the overall metric VQAScore even aligns better with human evaluation than the fine-grained evaluation metric DSG, a similar observation to (Li et al., 2024). Compared with these metrics, our metric *AlignScore* aligns with human evaluation much better, clearly demonstrating the effectiveness of our metric.

We would like to have some simple discussions about why other metrics do not work well on our benchmark. We attribute this problem to the complexity of our prompt and the natural drawbacks of these metrics. mGPT-CoT (Huang et al., 2023) often fails to follow instructions, making its provided evaluation score not reliable. CLIP Score utilizes vision-language models pretrained on general-purpose image-text pairs, lacking the capability of complex text understanding. The 3-in-1 score proposed in T2I-CompBench (Huang et al., 2023) utilizes a fixed text parser which relies on the specific structure of the prompt to work properly, making it hard to be adopted to any other benchmarks. VQAScore utilizes a large vision-language model to evaluate image-text alignment, however, current MLLMs still lack enough ability for accurately evaluating image-text alignment (Chen et al., 2024a), making its evaluation result not trustworthy enough. DSG is a QG/A method, which bears the same problems as we discussed before. Our metric directly assesses the alignment of each part following a structured format without introducing language models, thereby reducing the ambiguity of natural language and achieving the highest consistency with human evaluation.

Specifically, we present an example explaining why DSG fails to work properly. Consider a prompt "...1 laptop, 2 bowl. The first laptop is blue. The first bowl is brown. The second bowl is white." (a simplified real case), the generated questions using DSG are:

"... 7. Is the first bowl brown? 8. Is the second bowl white?"

VQA models face fundamental limitations in reliably distinguishing between ordinal references like "first bowl" and "second bowl" based solely on visual input and question text without any context. This inherent ambiguity naturally calls into question the reliability of such evaluation methodologies. Instead, our evaluation framework does not utilize language model, effectively resolving this problem.

## A.5 DISCUSSIONS ABOUT $AlignScore$

**Insights behind** $AlignScore$  The calculation of $AlignScore$ is based only on instance matching, while both color and relations are viewed as properties to be evaluated. This can address some interesting problems. We give an example as follows (which is simplified from real example):

Consider a prompt "A is black, on the left of B, on the right of C". If the generated result is a white A on the left of B and on the right of C, our evaluation metric will return $Acc = 0.67$, as it matches A in the prompt with A detected in the image. Therefore, it will view the color of A is incorrect ($\mathbb{I}[c_A = c'_A] = 0$) but the relations are correct ($\mathbb{I}[r_{AB} = r'_{AB}] = 1, \mathbb{I}[r_{AC} = r'_{AC}] = 1$), which aligns with our perception of this case.

However, decomposition-based methods face difficulties in handling such problem. For example, a QG/A method may generate a question "Is black A on the left of B?", which confuses color and relation, mistaking the correctly generated relation as incorrect.

Also, our instance matching evaluation successfully handles multiple instances belonging to the same category. We show another example (also a simplified on from a real example):

Consider a prompt "A1 is black, A2 is white". A QG/A method may generate such two questions "Is A black?" "Is A white?", which contradict with each other, making this evaluation completely meaningless. A similar example with the one mentioned before.

**The use of** $AlignScore$    We would like to discuss whether $AlignScore$ can be used beyond evaluating our benchmark. Though $AlignScore$ is designed to evaluate our benchmark, as long as the text prompt to evaluate only contains color and spatial relations without other properties, it can be organized into a structured format like ours and thus $AlignScore$ can be used for evaluation. Organizing the prompt into a structured format can be performed manually or with large language models like ChatGPT (OpenAI et al., 2023).

**Efficiency of** $AlignScore$ **Calculation**    Though we employ an exhaustive search for the best $f$, our calculation is still efficient since it only relies on relatively small models like CLIP and Mask2Former. In contrast, VQAScore, as an example, utilizes CLIP-FlanT5-XXL, which is extremely large, requiring much computation resources and inference time. Therefore, our $AlignScore$ is also computation resource and inference time efficient.

## B    EXPERIMENT DETAILS

### B.1    EXPERIMENT SETUP

We list detailed experiment setup in this part. First we list model checkpoints we used in Table 4.

Table 4: Details of our model weights.

| Model Name | Pretrained Weights |
| --- | --- |
| Stable-Diffusion-3 | Stable-Diffusion-3-Medium-Diffusers [1] |
| Stable-Diffusion-3.5-Medium | Stable-Diffusion-3.5-Medium [2] |
| Stable-Diffusion-3.5-Large-Turbo | Stable-Diffusion-3.5-Large-Turbo [3] |
| Stable-Diffusion-1.5 | Stable-Diffusion-v1-5 [4] |
| PixArt-$\Sigma$-XL | PixArt-Sigma-XL-2-1024-MS [5] |
| FLUX.1-dev | FLUX.1-dev [6] |

The inference hyper-parameters are listed in Table 5. $T$ is the total denoising steps and $w$ has the same meaning as in Equation 4 (except for Stable-Diffusion-3.5-Large-Turbo and FLUX.1-dev). These hyper-parameters are selected as their default settings.

All experiments are conducted on Nvidia A40 GPU. Inferencing our whole benchmark generally takes 6 to 10 GPU hours. Evaluating the generation result on our whole benchmark generally takes 1 GPU hour.

### B.2    MORE DISCUSSIONS ON EVALUATION RESULT

We make some more discussions about our evaluation results in this part.

**Evaluation Stability**    First of all, we would like to show that our benchmark provides stable evaluation results despite the randomness of sampling. We take Stable-Diffusion-3 and Pixart-$\Sigma$-XL as

---

[1] https://huggingface.co/stabilityai/stable-diffusion-3-medium-diffusers
[2] https://huggingface.co/stabilityai/stable-diffusion-3.5-medium
[3] https://huggingface.co/stabilityai/stable-diffusion-3.5-large-turbo
[4] https://huggingface.co/stable-diffusion-v1-5/stable-diffusion-v1-5
[5] https://huggingface.co/PixArt-alpha/PixArt-Sigma-XL-2-1024-MS
[6] https://huggingface.co/black-forest-labs/FLUX.1-dev

Table 5: Details of our inference hyper-parameter.

| Model Name | $T$ | $w$ |
|---|---|---|
| Stable-Diffusion-3 | 28 | 7.0 |
| Stable-Diffusion-3.5-Medium | 28 | 7.0 |
| Stable-Diffusion-3.5-Large-Turbo | 4 | - |
| Stable-Diffusion-1.5 | 50 | 7.5 |
| PixArt-$\Sigma$-XL | 20 | 4.5 |
| FLUX.1-dev | 28 | 3.5 |

2 examples and sample 4 times. We report the average and standard deviation of the metrics in Table 6.

Table 6: Average and standard deviation of the 4 evaluation results. $Acc$ results are multiplied by 100.

| Model Name | $Acc$ | $Bias$ |
|---|---|---|
| Stable-Diffusion-3 | $66.6_{0.18}$ | $1.53_{0.005}$ |
| Pixart-$\Sigma$-XL | $53.3_{0.18}$ | $2.24_{0.034}$ |

As can be seen from the results, the evaluation results bear a relatively small standard deviation, indicating that the results are quite stable under different sampling.

We would also like to note that this stability does not come from a lack of sampling diversity. Instead, we observe large diversity in different samples of the same prompt. For a single prompt, the model performance actually varies a lot! However, from the benchmark perspective, the model's performance remains quite stable, indicating that our relatively large benchmark indeed captures the model's performance instead of random noise, highlighting the importance of a large benchmark to ensure evaluation stability.

**Total Instance Numbers** The total number of object instances asked to generate is an important factor influencing the model's performance as mentioned. We investigate Stable-Diffusion-3.5-Medium and show its performance in Figure 7.

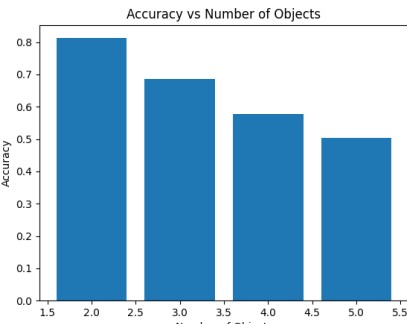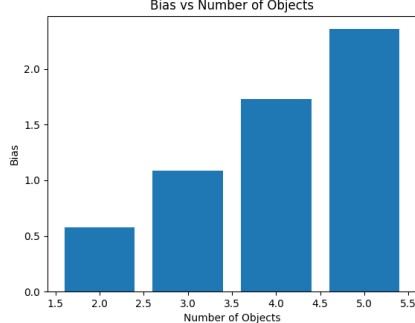

Figure 7: Model performance under different total number of object instances.

As can be seen, as the total instance number increases, model performance drops a lot, indicating current models still suffer from great challenges in dealing with complex prompts.

**The Randomness of Misalign** Unlike some idea that models are bad at generating specific content, we argue that, given a certain prompt, many failures in image-text alignment appear randomly. To be more detailed, if we view the overall evaluation results, there are indeed some contents that models generate worse than others. However, when given a certain prompt, the model can make

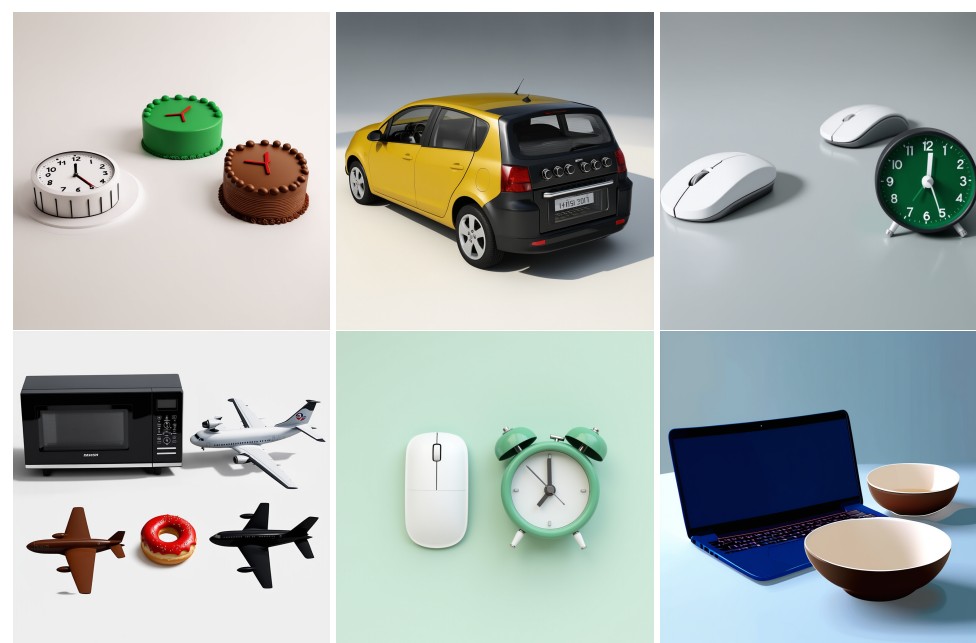

Figure 8: Misaligned cases.

alignment mistakes anywhere, and make different mistakes when different noise priors are sampled. This observation suggests that the performance of diffusion models highly depends on the sampled noise prior.

### B.3   CASE STUDY: REASONS OF MISALIGN

In this part, we would like to analyze some cases to investigate further how models fail on certain prompts. We would like to highlight a set of misaligned cases where the misaligned part comes from confusion with other parts of the prompt.

Previous works have pointed several types of this problem, but we would like to investigate deeper.

**Entity Leakage**   Entity leakage is objects disappearing or their outline breaking due to entities' positional fusion (Wu et al., 2024b; Wang et al., 2024). However, we would like to show that entity leakage can happen in a "gentle" way. As can be seen from the first image in Figure 8, the prompt is *"...3 clock, 1 cake ..."*. However, not only are the numbers of instances incorrect, but the generated cakes are clock-shaped! Though this is not "failure", as these generated instances are still "cakes", this is still a problem since the generated cakes seem unnatural, harming the quality of the generated image. We argue that this should also be viewed as a kind of entity leakage and should be properly addressed.

**Attribute Leakage**   Attribute leakage is an object generated with incorrect attributes and these attributes are actually attributed to other objects. We would like to show that attribute leakage can happen in another way. As can be seen from the second image in Figure 8, the prompt is *"...the car is yellow, the oven is black..."*, but the model generated a black and yellow car. Though most of the car is yellow, it still mixes an attribute that does not belong to it. This attribute leakage also harms image-text alignment.

**Number Leakage**   Number Leakage occurs when two categories are generated with incorrect numbers because they swapped their numbers. As can be seen from the third image in Figure 8, the prompt is *"...1 computer mouse, 2 clock"*, yet the model generates two computer mouses and one clock, exactly a swapped number. This number leakage highlights the problem that when mul-

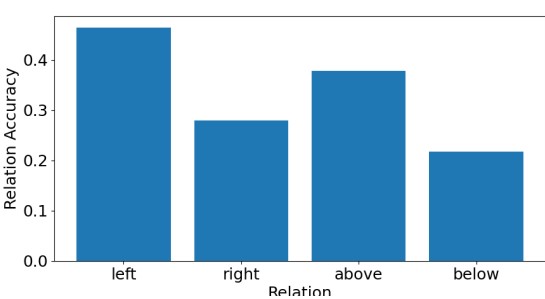

Figure 9: Accuracy of generating different relations on FLUX.1-dev.

tiple categories and multiple instances are introduced in the prompt, the model may not only mix their attributes, entities, but even their numbers, greatly harming image-text alignment.

**Entity Mixing**   Entity mixing is the case where the object instances of the same category mix their attributes. There are two types of entity mixing, the first of which is that the model not only mixes their attributes, but also merges the two instances into one single instance. As shown in the fifth image of Figure 8, where the prompt is *"... 1 computer mouse, 2 clock, ..., the first clock is white, ... the second clock is green, ..."*, however, the generated result only contains 1 clock with both white and green color, indicating a misalignment.

The second is that the model simply mixes their attributes. As can be seen in the sixth image in Figure 8, the prompt is *"... the first bowl is white, ... the second clock is white, ..."*. The model correctly generated two bowls, but both of them are brown and white colored, still a misalignment with the text description.

This entity mixing highlights the difficulty in generating multiple different instances of the same category, as models are very easily to be confused by multiple instances with the same category.

**Relation Bias**   Relation Bias is a phenomenon we observe that, the generated spatial relation is not only affected by relation specified in the prompt, but also affected by the order of instances appearing in the prompt. Models tend to put instances appearing earlier in the prompt on the top, left of the image. An example is shown in the fourth image in Figure 8, where the prompt is *"...The first airplane is white, on the right of the third airplane. The second airplane is brown, below the third airplane. The third airplane is black..."*. However, the white (first) airplane is above the black (third) airplane and the brown (second) airplane is on the left of the black (third) airplane, revealing failure in generating both relations. In fact, we calculate the accuracy of generated relations on FLUX.1-dev and observe the accuracy of generating "left" "above" is much higher than "right" "below", indicating a biased generation performance, as suggested in Figure 9.

### B.4   FAILURE CASE OF $AlignScore$

In this part, we discuss a failure case of $AlignScore$. We notice that sometimes the object detector may detect the same object instance twice, even after non-maximal suppression is applied, as the example in Figure 10.

As can be seen from the image, the clock on the right of the image is detected twice, which leads to incorrect $Bias$. Note that this does not affect $Acc$ because the color and relation result do not change. We argue that this is the inherent problem of the object detector. Since there can be multiple instances with the same category that are close to each other, decreasing the IoU threshold is not a good solution. Fortunately, according to our observation, this failure pattern does not occur often, and, after all, **as can be seen from the human evaluation results, our metric is still more trustworthy than previous metrics despite this potential failure**.

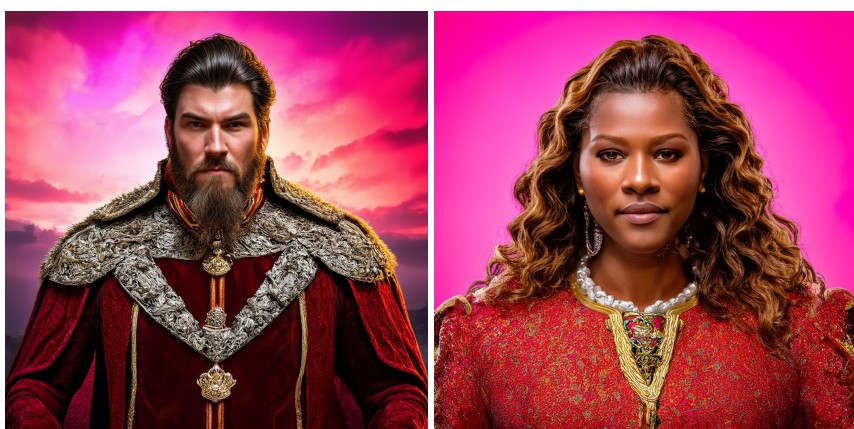

Figure 10: A failure case of $AlignScore$

Figure 11: Generated results using only $c_0$ (left) and using both $c_0$ and $c_1 - c_2$ (right).

## C  METHOD DISCUSSION

### C.1  ASSUMPTION VERIFICATION

We would like to verify our assumption that, when following formulation 6, the generation result will be a **semantic combination** of $c_0$ and $c_1 - c_2$. We generate an example with $c_0$ ="A photo of a king", $c_1$ ="woman", $c_2$ ="man" using Stable-Diffusion-3.

As can be seen, the generated result perfectly matches our assumption. The generated result with formulation 6 (right) is a **semantic combination** of $c_0$ and $c_1 - c_2$ ("queen") instead of **physical combination** of $c_0$ and $c_1 - c_2$ ("king and woman"). What's more, it has a similar style to using $c_0$ only, and the largest changes come from semantics. Therefore, we believe formulation 6 can be used to introduce more semantic information and better control generation with little influence on generation style.

### C.2  FURTHER DISCUSSION OF OUR METHOD

**Case Study**    We would like to present a case in Figure 12 to qualitatively show the performance of our method.

The prompt is: "A photo realistic image of 1 laptop, 2 bowl. The first laptop is blue. The first bowl is brown. The second bowl is white.". As can be seen, in the original image, only one brown bowl is generated. Therefore, our method formulate $c_1$ as "2 bowl. The second bowl is white" (the prompt that is not correctly generated) and $c_2$ as "1 bowl. The second bowl is not white" (the mistake in the current generation result).

As can be seen, with our method, the model correctly generates two bowls with correct colors, highlighting the effectiveness of our method. What's more, there are two interesting observations.

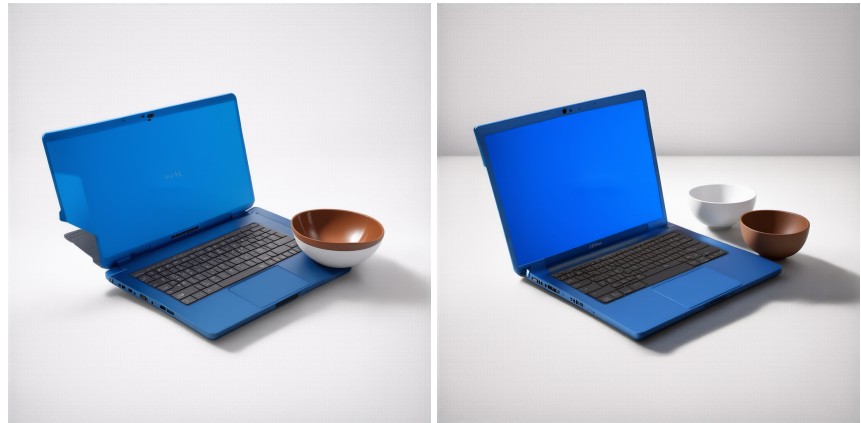

Figure 12: Example generated results w/o (left) and with (right) our method.

First of all, in the generation result without our method, the bowl is actually brown and white (though mostly brown), which is a sign of attribute leakage - the model fails to generate two bowls with separate attributes. Instead, it mixes the attributes and generates only one bowl with both attributes. However, after our method is applied, the model correctly generates two bowls and their attributes become separated - one pure brown and one pure white, meaning that our method successfully addresses the confusing parts of the prompt.

Secondly, in the generation result without our method, the edge of the laptop is flawed. It seems that there should be another object on the bottom left, where there is not. We suppose that the model intended to generate the second bowl there, but it failed. With our method, the model can better organize the positions of the objects, leading to more realistic generation results. As can be seen, the laptop in the right image has no flawed edges.

Thirdly, the generation result with our method bears a similar style to the original generated result, indicating that our method only affects generation content, but does not harm the generation diversity or generalization ability of the model, further suggesting the superiority of our method.

**Application of Our Method**  As discussed before, our method requires identifying the misaligned parts between the generated image and the text prompt, which is automatically done by the evaluation framework in our benchmark. In cases that our evaluation framework cannot be used, this can be done either manually or with other tools such as GPT-4 (OpenAI et al., 2023).

## D   LIMITATIONS

There are several limitations in this work. The prompts in our benchmark are constructed by templates, limiting the diversity of the prompts. Our evaluation metric relies on CLIP and object detector, making the trustworthiness of our metric depending on the performance of CLIP and the selected object detector. Our proposed method can only be applied after the image is generated and the misaligned parts between the generated image and text prompt is detected.

## E   ETHICAL CONSIDERATIONS

**Social Impact**  This work mainly presents a Text-to-Image benchmark constructed using MSCOCO object categories with evaluation metric and a method to improve image-text alignment, which are generally harmless and does not lead to negative social impacts.

**Human Evaluation**  Human evaluation is conducted on the basis of voluntary. Generally there are no potential harms of our human study since the prompt used to generate images does not describe any harmful scenarios. All annotators are well-educated grown-ups and are paid $20 per hour.

**Use of LLM**   We use LLM only for assist writing.

