# OpenReview forum: "M$^{3}$T2IBench: A Large-Scale Multi-Category, Multi-Instance, Multi-Relation Text-to-Image Benchmark"
_ICLR.cc/2026/Conference — Submitted to ICLR 2026_

### Official Review · Reviewer_EP4h · 2025-10-28

**Soundness:** 3
**Presentation:** 4
**Contribution:** 3
**Rating:** 4
**Confidence:** 4

**Summary:**

This paper addresses the poor alignment of Text-to-Image (T2I) models with complex prompts, particularly those with multiple instances of the same category. It introduces **M³T2IBench**, a large-scale (10,000-sample) benchmark for evaluating multi-category, multi-instance, and multi-relation generation. To evaluate this, the work proposes **AlignScore**, an object-detection-based metric that decomposes performance into instance counting ($Bias$) and attribute/relation accuracy ($Acc$), using an exhaustive search to map prompted and detected objects. Empirically, `AlignScore` achieves a high correlation with human judgment (Pearson $r=0.6711$), and results show current SOTA models perform poorly on this new benchmark. Finally, a training-free post-editing method, **Revise-Then-Enforce**, is proposed, demonstrating consistent alignment improvements across diffusion models.

**Strengths:**

1. The primary contribution, `M³T2IBench`, is well-motivated and addresses a significant, acknowledged gap in T2I evaluation: the difficulty with prompts containing multiple instances of the *same* category (e.g., "two bowls", "three clocks"), a feature missing from benchmarks like GenEval and T2I-CompBench (Table 1).
2. The proposed `AlignScore` metric demonstrates a significantly higher correlation with human judgment (Pearson $r=0.6711$) than established metrics like VQAScore ($0.4022$) and CLIPScore ($0.2296$). Its design, which separates `Bias` (count) from `Acc` (attribute/relation) and uses an optimal matching search, is methodologically sound for the target problem.
3. The `Revise-Then-Enforce` method is simple, training-free, and shown to be broadly effective, improving all tested open-source models (Table 2). The ablation study (Table 2, SD3 results) effectively justifies its design (Eq. 6) over simpler negative-prompt (Eq. 7) or positive-prompt (Eq. 8) variants.

**Weaknesses:**

1. The `AlignScore` metric is fundamentally dependent on the capabilities of its component tools: an object detector (Mask2Former) and a color classifier (CLIP). The authors acknowledge a failure mode where the detector double-counts an object (Figure 10), which directly impacts the `Bias` score. The robustness of the metric to such detector failures is not systematically evaluated.
2. The `AlignScore` calculation relies on an exhaustive search (Algorithm 2) to find the best matching $f$ between prompted and detected instances. The paper states this is "efficient," but provides no complexity analysis. This search space grows factorially with the number of instances, which may become computationally infeasible for prompts more complex than those in the benchmark (max 5 instances of the same category).
3. The proposed `Revise-Then-Enforce` method is a post-editing technique, requiring a full generation pass *before* applying the correction. This doubles the inference cost (one pass to find errors, one to correct). This practical limitation is not discussed in the main paper.
4. The benchmark's prompts are generated from rigid templates (Section 3.1), which, while ensuring quality and avoiding LLM subjectivity, limits linguistic diversity. This is acknowledged as a limitation but is a notable tradeoff, as models may overfit to this specific templated sentence structure.

**Questions:**

1. Regarding the exhaustive search in Algorithm 2 for `AlignScore`: What is the computational cost of this step as the number of instances ($N$) and detected objects ($M$) increases? Could this step be replaced with a more efficient optimal transport or bipartite matching algorithm, and what would be the impact on the final score?
2. The `Revise-Then-Enforce` method (Section 4.2) requires constructing paired prompts $c_1$ (what should be) and $c_2$ (what is wrong). In the case study (Figure 12), $c_2$ is "1 bowl. The second bowl is not white." How sensitive is the method's performance to the precise phrasing of $c_2$? For example, would "1 brown bowl" or simply "1 bowl" have worked as well?
3. The analysis in Figure 9 suggests a "Relation Bias," where models are better at "left" and "above" than "right" and "below". Does this bias correlate with the order of object mentions in the prompt templates? Could this be an artifact of the template-based prompt construction rather than an inherent model bias?
4. The detector's failure case in Figure 10 shows one clock detected twice, which would incorrectly increase the `Bias` score. How prevalent is this duplicate-detection failure in the 10,000-sample benchmark, and does it significantly skew the reported `Bias` results for the models in Table 2?

---

> ### Author Response · Authors · 2025-11-18
>
> We sincerely appreciate the time and effort you have dedicated to reviewing our manuscript. Below, we address your questions and provide clarifications as follows:
>
> > The `AlignScore` metric is fundamentally dependent on the capabilities of its component tools: an object detector (Mask2Former) and a color classifier (CLIP). The authors acknowledge a failure mode where the detector double-counts an object (Figure 10), which directly impacts the `Bias` score. The robustness of the metric to such detector failures is not systematically evaluated.
> >
> > The detector's failure case in Figure 10 shows one clock detected twice, which would incorrectly increase the `Bias` score. How prevalent is this duplicate-detection failure in the 10,000-sample benchmark, and does it significantly skew the reported `Bias` results for the models in Table 2?
>
> The automatic detectors employed are the same established tools used in GenEval [1], a benchmark whose human validation has already been established and whose detectors are generally considered trustworthy within the field. We performed internal observations that led to similar conclusions regarding their reliability.
>
> To mitigate the specific issue of duplicate detection, we have carefully implemented Non-Maximum Suppression (NMS) during the detection phase. While we must admit that isolated failure cases, such as the one illustrated in Figure 10, do still occur, these instances are not sufficiently prevalent to significantly skew the overall reported performance or model rankings.
>
> To further validate the trustworthiness of our final results, we directly calculated the correlation between our automatic evaluation scores and human evaluation (reported in Table 3 of Appendix A). The high correlation found confirms that, despite the presence of occasional detector failures, our final evaluation metric is generally trustworthy and reliably reflects human judgment.
>
> > The `AlignScore` calculation relies on an exhaustive search (Algorithm 2) to find the best matching  between prompted and detected instances. The paper states this is "efficient," but provides no complexity analysis. This search space grows factorially with the number of instances, which may become computationally infeasible for prompts more complex than those in the benchmark (max 5 instances of the same category).
> >
> > Regarding the exhaustive search in Algorithm 2 for `AlignScore`: What is the computational cost of this step as the number of instances (N) and detected objects (M) increases? Could this step be replaced with a more efficient optimal transport or bipartite matching algorithm, and what would be the impact on the final score?
>
> We appreciate this critical question regarding the complexity of the exhaustive search in Algorithm 2. Firstly, we would like to clarify that the term "efficient" was used as an empirical observation relative to other metrics like VQAScore. For instance, evaluating our full benchmark takes only about one hour using AlignScore, compared to several hours for VQAScore, due to the reliance on much larger models in the latter.
>
> We acknowledge that the search space of the exhaustive search does grow factorially, with an approximate complexity of $O\left(P_{\min(N,M)}^{\max(N,M)}\right)$ (where $P$ denotes the permutation function). However, even for a challenging case with $N=10$ prompted instances and $M=10$ detected objects of the same category, the time consumed by the exhaustive search is still less than 1 second. This runtime is negligible compared with inferencing with larger models. Furthermore, if an image contains significantly more instances, the primary limiting factor would shift to the **generation model's capability** to handle such complexity, which lies beyond the application boundary of our matching algorithm.
>
> Regarding alternative algorithms, while replacing the exhaustive search with a more efficient optimal transport or bipartite matching algorithm is conceptually possible, our consultations with experts did not yield a simple and suitable solution that would maintain the required alignment semantics. While searching and matching with some randomness might be more feasible, they risk introducing additional matching errors. Given that our current method performs well within a reasonable number of instances (up to $5$ in our benchmark), we believe the exploration of more efficient matching algorithms is best left for future work.
>
> Thank you for raising this important issue; we will add a detailed discussion of the complexity and efficiency.

---

> > ### Author Response · Authors · 2025-11-18
> >
> > > The proposed `Revise-Then-Enforce` method is a post-editing technique, requiring a full generation pass *before* applying the correction. This doubles the inference cost (one pass to find errors, one to correct). This practical limitation is not discussed in the main paper.
> >
> > Thank you for raising this concern, we acknowledge this as one of the limitations of our work and have discussed it briefly in Appendix D. We will add more clear discussions about this issue.
> >
> > > The benchmark's prompts are generated from rigid templates (Section 3.1), which, while ensuring quality and avoiding LLM subjectivity, limits linguistic diversity. This is acknowledged as a limitation but is a notable tradeoff, as models may overfit to this specific templated sentence structure.
> >
> > We appreciate the concern regarding the potential for models to "overfit" to the templated sentence structure. However, we would like to clarify two main points:
> >
> > Firstly, our benchmark is designed strictly for evaluation, not for model training. Therefore, the risk of traditional overfitting during training*is not applicable here.
> >
> > As for evaluation, the phenomenon of models tailoring performance to specific evaluation criteria, or "metric hacking," is a universal challenge across all benchmarks instead of our benchmark only (e.g., overfitting to sentence structure, style, length, or specific word choices). And we would like to emphasize that the primary goal of our template-based approach is to rigorously and objectively evaluate T2I models' foundational ability to handle complex prompts involving multiple object instances and relations. Our benchmark provides a valuable, standardized resource for diagnosing these specific compositional failures. We agree that reliance on a single benchmark is unwise, and our work should be viewed as a complementary diagnostic tool alongside broader, more diverse evaluation sets.
> >
> > > The `Revise-Then-Enforce` method (Section 4.2) requires constructing paired prompts $c_1$  (what should be) and $c_2$  (what is wrong). In the case study (Figure 12),  is "1 bowl. The second bowl is not white." How sensitive is the method's performance to the precise phrasing of $c_2$? For example, would "1 brown bowl" or simply "1 bowl" have worked as well?
> >
> > We did not perform a detailed analysis of the precise phrasing of $c_2$ because our intention was to make the Revise-Then-Enforce method fully automatic in its implementation. Consequently, we phrase $c_2$ in a rule-based manner by strictly comparing the automatic evaluation result against the original structured prompt.
> >
> > We acknowledge that this programmatic formulation may not be linguistically optimal, yet it already yields satisfactory results as demonstrated in our experiments. We agree that exploring variations in $c_2$ phrasing—for example, by using "1 brown bowl" or other more natural language constructions—would likely produce even better performance, and we believe this is a valuable direction to be left for future work.
> >
> > > The analysis in Figure 9 suggests a "Relation Bias," where models are better at "left" and "above" than "right" and "below". Does this bias correlate with the order of object mentions in the prompt templates? Could this be an artifact of the template-based prompt construction rather than an inherent model bias?
> >
> > We appreciate this valuable observation. This is actually a really interesting part during our evaluation.
> >
> > First, we wish to clarify that the observed "Relation Bias," as stated in Lines 1112–1115, **does not simply mean models are better at "left" and "above."** Instead, it refers to the tendency that **"Models tend to put instances appearing earlier in the prompt on the top, left of the image."**
> >
> > Therefore, the higher accuracy observed for the "left" and "above" relations is actually a direct result of **an inherent model bias interacting with our template design.** Our templates tend to introduce object relations immediately after an instance is mentioned. Since models already possess a bias to prioritize and place early-mentioned objects toward the top-left of the image, they naturally perform better when the prompt requires a relation (like "left" or "above") that is consistent with this inherent bias. Conversely, they perform worse when the prompt requires a relation (like "right" or "below") that contradicts this bias.
> >
> > In summary, **this phenomenon is an inherent model bias, and our template design effectively helped to reveal and quantify this specific problem.** This bias is not merely an artifact caused by the template construction itself.
> >
> > We again sincerely appreciate the time and careful consideration you have given to reviewing our manuscript. We hope our responses have adequately addressed your questions and concerns. Should any additional clarifications be needed, we would be happy to provide further information.

---

> > > ### Comment · Reviewer_EP4h · 2025-11-27
> > >
> > > Thank you for the comprehensive response from the author, I have decided to improve my score.

---

### Official Review · Reviewer_ShuA · 2025-10-31

**Soundness:** 2
**Presentation:** 3
**Contribution:** 2
**Rating:** 2
**Confidence:** 4

**Summary:**

The authors introduce M3T2IBench, a text-to-image benchmark featuring multiple categories, instances, and relations, along with AlignScore, a metric that correlates better with human judgment. Their evaluation of various T2I models exposes difficulties in generating images that align well with complex prompts. To address this, they present a training-free "Revise-Then-Enforce" post-editing approach that improves alignment between generated images and text prompts.

**Strengths:**

1. The paper proposes to evaluate both acc and bias, with an exhaustive searching method to determine the final acc. The idea of evaluating two metrics is reasonable.

2. The paper is well-written and easy to follow.

**Weaknesses:**

1. Lack of novelty of the proposed revise-then-enforce method. First, the method appears to be hardly related to the proposed benchmark and scoring metrics. This makes the paper separate into two unrelated parts. Additionally, a similar idea has already been explored in earlier works, but the paper fails to mention [1].
2. In Line 201, the benchmark refrains from mentioning words like fresh and majestically to ensure accurate evaluation. This idea does not seem reasonable and therefore severely limits the scope of testing for the benchmark. A holistic benchmark should encompass all the frequently mentioned words by users, rather than focusing on words that are easy to judge and discarding the subjective ones. There are far more attributes that often appear in the prompts, such as moods and shapes.
3. Besides, the object-centered design seems to limit the comprehensiveness of the benchmark. There are aspects more than objects, like complex relations and styles.
4. The weakness of QG/A is not very clear. Why can this method not correctly deal with multiple instances belonging to the same category in the prompt?

[1] Wang, Zihao, et al. "Concept algebra for (score-based) text-controlled generative models." Advances in Neural Information Processing Systems 36 (2023): 35331-35349.

**Questions:**

1. Clarify the connection between the proposed method and [1].

2. What is the relation between the proposed benchmark and method?

3. More clarification and examples of the limit of QG/A methods.

Please also refer to the weakness part.

---

> ### Author Response · Authors · 2025-11-18
>
> We sincerely appreciate the time and effort you have dedicated to reviewing our manuscript. Below, we address your questions and provide clarifications as follows:
>
> > Lack of novelty of the proposed revise-then-enforce method. First, the method appears to be hardly related to the proposed benchmark and scoring metrics. This makes the paper separate into two unrelated parts. Additionally, a similar idea has already been explored in earlier works, but the paper fails to mention [1].
>
> Thank you for raising these important points.
>
> First, regarding the relationship between the method and the benchmark: the revise-then-enforce method is proposed as a proof-of-concept solution to demonstrate a potential direction for improving model performance on the specific challenges revealed by our benchmark. Its inclusion is intended to provide potential guidance for future work on addressing the shortcomings we identified, thereby connecting the problem (benchmark) with a potential solution (method).
>
> Second, regarding novelty and prior work: as we discussed in Lines 355–368, similar ideas have been proposed previously. However, we show empirically that our specific formulation of the revise-then-enforce method provides the best result among the related approaches we tested (demonstrated in Table 2). We would like to note that the method proposed in [1] bears a closer resemblance to Equation (7) (one of a baseline settings) in our paper. We will ensure this specific connection is added to the discussion accordingly. Thank you again for pointing out this highly relevant reference.
>
> > In Line 201, the benchmark refrains from mentioning words like fresh and majestically to ensure accurate evaluation. This idea does not seem reasonable and therefore severely limits the scope of testing for the benchmark. A holistic benchmark should encompass all the frequently mentioned words by users, rather than focusing on words that are easy to judge and discarding the subjective ones. There are far more attributes that often appear in the prompts, such as moods and shapes.
> >
> > Besides, the object-centered design seems to limit the comprehensiveness of the benchmark. There are aspects more than objects, like complex relations and styles.
>
> Firstly, our decision to exclude subjective terms (like 'fresh' or 'majestic') and focus on verifiable attributes follows the established methodology of GenEval [2]. This approach is crucial for enabling a trustworthy and accurate automatic evaluation using current detection tools.
>
> Secondly, we argue that the **main focus of our research lies in rigorously evaluating a model's ability to handle multiple object instances and compositional correctness within a single, complex scene**. As demonstrated in Table 2, even this limited, focused evaluation already poses significant challenges to existing T2I models, underscoring the immediate value and necessity of our proposed benchmark.
>
> While we agree that exploring a wider range of subjective attributes, moods, shapes, complex relations, and styles is important for a comprehensive evaluation, these aspects of prompt complexity can be more appropriately left for future work once the core compositional failure modes identified by our benchmark are addressed.

---

> > ### Author Response · Authors · 2025-11-18
> >
> > > The weakness of QG/A is not very clear. Why can this method not correctly deal with multiple instances belonging to the same category in the prompt?
> >
> > The core weakness of Question Generation/Answering (QG/A) methods, as discussed in Appendix A.4 and A.5, is their inability to generate reasonable and unambiguous questions when faced with multiple instances of the same object category in a prompt. It is non-trivial for these methods to reliably distinguish between identical instances using only textual questions.
> >
> > For instance, consider the prompt described in **Lines 894-898**:
> >
> > *"$\ldots$ 1 laptop, 2 bowl. The first laptop is blue. The first bowl is brown. The second bowl is white."*
> >
> > A popular QG/A method, such as DSG, might generate questions like: *"7. Is the first bowl brown? 8. Is the second bowl white?"*
> >
> > When a Visual Question Answering (VQA) model is presented with the generated image and such a question, it is unable to distinguish which object instance in the image corresponds to "the first bowl" or "the second bowl." The questions are therefore **unanswerable** based solely on the image and the question text. QG/A methods frequently generate these ambiguous, unanswerable questions because they lack a robust mechanism to uniquely identify and refer to instances belonging to the same category within the generated textual query.
> >
> > We again sincerely appreciate the time and careful consideration you have given to reviewing our manuscript. We hope our responses have adequately addressed your questions and concerns. Should any additional clarifications be needed, we would be happy to provide further information.
> >
> > [1] Wang, Zihao, et al. "Concept algebra for (score-based) text-controlled generative models." Advances in Neural Information Processing Systems 36 (2023): 35331-35349.
> >
> > [2] Dhruba Ghosh, Hannaneh Hajishirzi, and Ludwig Schmidt. Geneval: An object-focused framework for evaluating text-to-image alignment. In A. Oh, T. Naumann, A. Globerson, K. Saenko, M. Hardt, and S. Levine (eds.), Advances in Neural Information Processing Systems, volume 36, pp. 52132–52152. Curran Associates, Inc., 2023.

---

### Official Review · Reviewer_WmJY · 2025-11-01

**Soundness:** 3
**Presentation:** 3
**Contribution:** 2
**Rating:** 4
**Confidence:** 4

**Summary:**

This paper introduces M3T2IBench, a large-scale benchmark designed to evaluate text-to-image models on complex compositional prompts. It contains 10,000 structured prompts, which is significantly larger and more diverse than prior benchmarks. The prompts include multiple objects, categories, attributes, and spatial relations, allowing for rigorous evaluation of multi-instance and relational reasoning. The benchmark uses automated metrics to separately measure bias (errors in the number of generated instances) and accuracy (correctness of attributes and relations), relying on object detection and color classification systems. Experiments show that state-of-the-art generators—including Stable Diffusion 3, FLUX, and DALL-E 3—perform poorly, with accuracy below 70% and consistent object count errors. These results highlight the gap between current T2I model capabilities and real compositional understanding.

**Strengths:**

1. Significant Benchmark Scale and Complexity: It presents largest T2I compositional alignment dataset to date (10k prompts), and supports long prompts, many relations, and multiple instances per category

2. Fine-Grained and Structured Metric Design. The paper evaluates object counts, colors, spatial relations via automated detectors, and distinguishes between bias (count errors) and accuracy (attributes/relations)

**Weaknesses:**

The benchmark depends heavily on automatic detectors for object, color, and relation evaluation. While this enables scalability, it also inherits all failure cases of those detectors, particularly in stylized or abstract generations. The lack of reported human validation raises questions about the accuracy of the scores and whether false detector errors might inflate failure rates.

Another limitation is the synthetic nature of prompt construction. Prompts are generated using probabilistic rules for relations and attributes, which may not reflect real-world usage patterns. As a result, while the benchmark is rigorous, it may evaluate models on scenarios that are technically difficult but less representative of natural user intent.

The paper also does not deeply analyze failure modes or propose guidance for model improvement beyond brief mentions of an included approach. More detailed discussion or qualitative insight into model breakdowns would strengthen the contribution. Finally, the benchmark over-emphasizes structural correctness, which means it may undervalue creative fidelity or semantic coherence compared to strict compositional rule-following.

**Questions:**

none

---

> ### Author Response · Authors · 2025-11-18
>
> We sincerely appreciate the time and effort you have dedicated to reviewing our manuscript. Below, we address your questions and provide clarifications as follows:
>
> > The benchmark depends heavily on automatic detectors for object, color, and relation evaluation. While this enables scalability, it also inherits all failure cases of those detectors, particularly in stylized or abstract generations. The lack of reported human validation raises questions about the accuracy of the scores and whether false detector errors might inflate failure rates.
>
> The detectors we employ are identical to those used in the established GenEval benchmark [1], where their human validation and general trustworthiness have already been demonstrated. Furthermore, to specifically validate the accuracy of our resulting evaluation scores, we conducted a direct comparison between our automatic metrics and a set of human evaluations. The correlation results (reported in Table 3 of Appendix A) show that our automatic evaluation, utilizing these tools, aligns strongly with human judgment, confirming the overall reliability of our reported scores.
>
> > Another limitation is the synthetic nature of prompt construction. Prompts are generated using probabilistic rules for relations and attributes, which may not reflect real-world usage patterns. As a result, while the benchmark is rigorous, it may evaluate models on scenarios that are technically difficult but less representative of natural user intent.
>
> We acknowledge that the synthetic nature of our prompt construction is a limitation, as discussed in Appendix D. However, the primary goal of our benchmark is to rigorously evaluate a model's capability in handling complex and compositional instructions. We specifically designed these prompts to test the limits of T2I models by creating demanding, structured scenarios. We believe that our proposed benchmark successfully and systematically revealed current shortcomings in existing T2I models' ability to follow such complex commands, which underscores its practical value for diagnostic evaluation and targeted model improvement.
>
> > The paper also does not deeply analyze failure modes or propose guidance for model improvement beyond brief mentions of an included approach. More detailed discussion or qualitative insight into model breakdowns would strengthen the contribution.
>
> We appreciate this suggestion. Due to the space constraints of the main paper, we included the detailed analysis of failure modes—covering both quantitative and qualitative insights into model breakdowns—in **Appendix B.3**. We will ensure this is clearly highlighted and referenced within the main body of the paper to guide the reader to this important discussion.
>
> > Finally, the benchmark over-emphasizes structural correctness, which means it may undervalue creative fidelity or semantic coherence compared to strict compositional rule-following.
>
> We acknowledge that our benchmark heavily emphasizes structural correctness and strict compositional rule-following. This design choice is deliberate, as the **primary goal of our work is to rigorously evaluate the instruction-following capability** of existing Text-to-Image (T2I) models when presented with complex, multi-instance prompts.
>
> Creative fidelity, while a valuable characteristic, falls outside our discussion scope of evaluating a T2I model's ability to accurately follow instructions. Furthermore, our results clearly demonstrate that existing models frequently fail even on strict compositional rule-following. Given this current performance gap, the challenging discussion regarding the more complex requirement of semantic coherence is a vital area that we believe is best left for future work.
>
> We again sincerely appreciate the time and careful consideration you have given to reviewing our manuscript. We hope our responses have adequately addressed your questions and concerns. Should any additional clarifications be needed, we would be happy to provide further information.
>
> [1] Dhruba Ghosh, Hannaneh Hajishirzi, and Ludwig Schmidt. Geneval: An object-focused framework for evaluating text-to-image alignment. In A. Oh, T. Naumann, A. Globerson, K. Saenko, M. Hardt, and S. Levine (eds.), Advances in Neural Information Processing Systems, volume 36, pp. 52132–52152. Curran Associates, Inc., 2023.

---

### Official Review · Reviewer_HPHY · 2025-11-11

**Soundness:** 4
**Presentation:** 4
**Contribution:** 3
**Rating:** 8
**Confidence:** 3

**Summary:**

The paper introduces M3T2IBench, a new large-scale, multi-category, multi-instance, multi-relation benchmark. It focuses on prompts that have many objects, repeated categories, colors, and spatial relations. The benchmark comes with an automatic metric called AlignScore which combines two parts (1) Bias for counting category mistakes and (2) Accuracy for checking if attributes and relations are correct after matching each text object to a visual object. The paper shows that AlignScore correlates better with human judgement than alternatives. It also shows that popular models struggle as prompt complexity increases. The authors propose Revise and Enforce that edits the prompt using what went wrong and pushes generation toward the failed parts without retraining helping improving alignment across several diffusion models.

**Strengths:**

The paper’s original contribution focuses on real world scenarios which other metrics lack. The prompt construction method is simple, general and easily applicable across diffusion models, and needs no retraining. The metrics are defined clearly and human ratings align better with AlignScore compared to alternatives. The paper is easy to read and the information flows logically. The paper’s contribution lies in coming up with a structured large-scale benchmark making it valuable for real world use cases and the gains from R&E method is consistent and model-agnostic.

**Weaknesses:**

In the paper, the attributes used are mostly focused on color. Adding additional attributes like shape can help make accuracy apply to real world use cases and not just color and position. DALLE-3 comparison has only 100 prompts which is relatively smaller. Revise and Enforce is assumed to depend on correctly identifying the failed parts. It would be great to explain the method used to identify the failed parts and turn them into prompts.

**Questions:**

1. How much will the scores change if the detector or color classifier is changed? Any plan to ensemble or calibrate per category?
2. If you change the balance between Bias and Accuracy in AlignScore, do model rankings stay stable relative to human judgments
3. Can you describe in more details about the post editing method from detection errors to two edited prompts?

---

> ### Author Response · Authors · 2025-11-18
>
> We sincerely appreciate the time and effort you have dedicated to reviewing our manuscript. Below, we address your questions and provide clarifications as follows:
>
> > How much will the scores change if the detector or color classifier is changed? Any plan to ensemble or calibrate per category?
>
> If a stronger detector and color classifier were utilized, the resulting computed scores would naturally exhibit greater accuracy. However, in this work, we directly follow the implementation and standard configuration established by GenEval [1]. The detector and color classifier currently employed are widely recognized and considered generally trustworthy within this evaluation framework.
>
> > If you change the balance between Bias and Accuracy in AlignScore, do model rankings stay stable relative to human judgments?
>
> We explored this by testing a generalized formulation of AlignScore: $AlignScore = \dfrac{1}{1+\alpha}(Acc + \dfrac{\alpha}{Bias+1})$, and varying the parameter $\alpha$. Determining the single **optimal $\alpha$** is challenging, as the ideal balance between accuracy and bias may vary based on different user or application requirements.
>
> To select a robust setting, we calculated the correlation between the evaluation results (using different $\alpha$ values) and our annotated human judgments. We found that the current formulation used in the paper, corresponding to $\alpha=1$, provides satisfactory results. While other values, such as $\alpha=2$, also yielded reasonable correlations, we chose to maintain $\alpha=1$ as our final setup due to its simplicity and balance.
>
> > Can you describe in more details about the post editing method from detection errors to two edited prompts?
>
> Just as illustrated in the example provided in Appendix C.2, if the original prompt is structured as, for instance, "... 2 bowl ... The second bowl is white," and the evaluation framework detects that the generated result contains only a brown bowl (detected), we construct two edited prompts:
>
> **$c_1$ (Intended):** This prompt captures how the image **should have been generated**. It includes the specific parts of the original prompt that were violated, such as: "2 bowl, the second bowl is white."
>
> **$c_2$ (Mistaken):** This prompt describes how the image was **mistakenly generated**. It is formed by comparing the structured evaluation result against the original intended prompt structure, resulting in a description like: "1 bowl, the second bowl is not white."
>
> We again sincerely appreciate the time and careful consideration you have given to reviewing our manuscript. We hope our responses have adequately addressed your questions and concerns. Should any additional clarifications be needed, we would be happy to provide further information.
>
> [1] Dhruba Ghosh, Hannaneh Hajishirzi, and Ludwig Schmidt. Geneval: An object-focused framework for evaluating text-to-image alignment. In A. Oh, T. Naumann, A. Globerson, K. Saenko, M. Hardt, and S. Levine (eds.), Advances in Neural Information Processing Systems, volume 36, pp. 52132–52152. Curran Associates, Inc., 2023.

---

### Meta-Review · Area_Chair_4yfB · 2026-01-07

**Summary:**

The scores of this paper are 8,4,2,4. The reviewer R4 indicated they would improve the rating. So this paper has mixed opinions.

The paper's contribution is two-fold. One is an evaluation metric and benchmark for t2i models. The second is an improved t2i method which is training free.

The AC went through the review comments and discussions and thinks that this paper has the following limitations.

- Reviewers asked about detector robustness. The authors replied and said that they used GenEval detectors. AC thinks that this is not sufficient and encourage authors to try more detector variants. This is because the GenEval detectors are known to have some bias which hinders model evaluation. Authors are encouraged to look deeper.

- Reviewers also asked about the way prompts are created. The current way may overlook important aspects like shape. Authors replied that simpler prompts leave less leeway for the LLM to make mistakes. AC thinks that authors should try stronger LLMs and more complex and diverse prompts.

- The revise-then-enforce method is a bit detached from the evaluation/metric part. It looks like a different work. Also, authors are encouraged to compare this method with existing training-free methods like Attend-and-Excite and perhaps more recent ones.

Based on the above considerations, AC recommends reject.

**Reviewer Concerns:**

See above.

**Reviewer Scores:**

See above.

---

### Decision · Program_Chairs · 2026-01-26

Reject